# Increasing calling accuracy, coverage, and read-depth in sequence data by the use of haplotype blocks

Torsten Pook[1¤]*, Adnane Nemri[2], Eric Gerardo Gonzalez Segovia[3], Daniel Valle Torres[3], Henner Simianer[1], Chris-Carolin Schoen[3]

**1** Center for Integrated Breeding Research, Animal Breeding and Genetics Group, University of Goettingen, Goettingen, Germany, **2** KWS SAAT SE & Co. KGaA, Einbeck, Germany, **3** Plant Breeding, Technical University of Munich, TUM School of Life Sciences Weihenstephan, Freising, Germany

¤ Current address: Animal Breeding and Genetics, University of Goettingen, Goettingen, Germany
* torsten.pook@uni-goettingen.de

**Data Availability Statement:** Genomic data for chromosome 1 of the 321 DH-lines that was generated via sequencing with 0.5X read-depth after preprocessing in FreeBayes is available at

## Abstract

High-throughput genotyping of large numbers of lines remains a key challenge in plant genetics, requiring geneticists and breeders to find a balance between data quality and the number of genotyped lines under a variety of different existing genotyping technologies when resources are limited. In this work, we are proposing a new imputation pipeline ("HBimpute") that can be used to generate high-quality genomic data from low read-depth whole-genome-sequence data. The key idea of the pipeline is the use of haplotype blocks from the software HaploBlocker to identify locally similar lines and subsequently use the reads of all locally similar lines in the variant calling for a specific line. The effectiveness of the pipeline is showcased on a dataset of 321 doubled haploid lines of a European maize landrace, which were sequenced at 0.5X read-depth. The overall imputing error rates are cut in half compared to state-of-the-art software like BEAGLE and STITCH, while the average read-depth is increased to 83X, thus enabling the calling of copy number variation. The usefulness of the obtained imputed data panel is further evaluated by comparing the performance of sequence data in common breeding applications to that of genomic data generated with a genotyping array. For both genome-wide association studies and genomic prediction, results are on par or even slightly better than results obtained with high-density array data (600k). In particular for genomic prediction, we observe slightly higher data quality for the sequence data compared to the 600k array in the form of higher prediction accuracies. This occurred specifically when reducing the data panel to the set of overlapping markers between sequence and array, indicating that sequencing data can benefit from the same marker ascertainment as used in the array process to increase the quality and usability of genomic data.

https://github.com/tpook92/HBimpute. Genomic data for chromosome 1 for the 321 DH-lines that was generated via the 600k Affymetrix Axiom Maize Genotyping Array is available at https://github.com/tpook92/HaploBlocker. Genomic data for the other chromosomes and raw data are available upon request. All source code underlying the HBimpute step is provided via GitHub (https://github.com/tpook92/HBimpute) and implemented in the associated R-package HBimpute. Scripts to run the HBimpute pipeline and perform tests for downstream analysis (error rates, GWAS and GP) are also available at https://github.com/tpook92/HBimpute/. All genetic data is jointly owned by KWS SAAT SE & Co. KGaA, Technical University of Munich, and University of Hohenheim. Therefore, we have no permission to share all raw data. However, the uploaded data should easily be sufficient to validate the method proposed in our manuscript. Contact regarding data questions is Ulrike Utans-Schneitz (utansschneitz@tum.de).

**Funding:** TP, EGGS, CCS, HS received financial support from the German Federal Ministry of Education and Research (BMBF, https://www.bmbf.de/) via the project MAZE - "Accessing the genomic and functional diversity of maize to improve quantitative traits"; Funding ID: 031B0882). We acknowledge support by the Open Access Publication Funds of the Göttingen University. The funders had no role in study design, data collection and analysis, decision to publish, or preparation of the manuscript.

**Competing interests:** I have read the journal's policy and the authors of this manuscript have the following competing interests: The presented HBimpute step is patent pending under application number EP20201121.9. Patent applicants are KWS SAAT SE & Co. KGaA and the University of Goettingen. Inventors are Torsten Pook and Adnane Nemri. Use in academia is possible without restrictions.

## Author summary

High-throughput genotyping of large numbers of lines remains a key challenge in plant genetics and breeding. Cost, precision, and throughput must be balanced to achieve optimal efficiency given available technologies and finite resources. Although genotyping arrays are still considered the gold standard in high-throughput quantitative genetics, recent advances in sequencing provide new opportunities. Both the quality and cost of genomic data generated based on sequencing are highly dependent on the used read-depth. In this work, we propose a new imputation pipeline ("HBimpute") that uses haplotype blocks to detect individuals of the same genetic origin and subsequently uses all reads of those individuals in the variant calling. Thus, the obtained virtual read-depth is artificially increased, leading to higher calling accuracy, coverage, and the ability to call copy number variation based on low read-depth sequencing data. To conclude, our approach makes sequencing a cost-competitive alternative to genotyping arrays with the added benefit of allowing the calling of structural variation.

## Introduction

High-throughput genotyping of large numbers of lines remains a key challenge in plant genetics and breeding. Cost, precision, and throughput must be balanced to achieve optimal efficiencies given available genotyping technologies and finite resources. Improvements in the cost-effectiveness or resolution of high-throughput genotyping are a worthwhile goal to support efforts from breeders to increase genetic gain and thereby aid in feeding the world's rapidly growing human population [1].

As of today, high-throughput genotyping is commonly performed using single nucleotide polymorphism (SNP) arrays in most common crops and livestock species. Genotyping arrays can have various marker densities, ranging from 10k SNPs [2] to 50k [3, 4] to 600k SNPs [3, 5, 6], are relatively straightforward to use [7], and typically produce robust genomic data with relatively few missing calls or calling errors [6]. As a result, genotyping arrays are widely used for a broad range of applications, including diversity analysis [8, 9], genomic selection [10, 11] or genome-wide association studies [12, 13]. Limitations of the technology comprise the complexity and cost of designing the arrays, their inability of typing *de novo* polymorphisms, and their lack of flexibility in the choice of marker positions. In addition, array markers are typically SNPs selected to be in relatively conserved regions of the genome [14, 15], i.e. by design they provide little information on structural variants, although calling of structural variation, in principle, is also possible via genotyping arrays [16].

In recent years, rapid advances in next-generation sequencing (NGS) have enabled targeted genotyping-by-sequencing (GBS) and whole-genome-sequencing (WGS) to become cheaper, more accurate, and widely available [17, 18]. Compared to genotyping arrays, GBS and WGS data provide additional information such as the local read-depth and a higher overall marker density, which have been successfully used in a variety of studies [19–21]. Studies that use GBS or WGS data to call structural variation typically use a read-depth of at least 5X [22, 23]. For applications such as genomic prediction, the use of 1X to 2X read-depth would be imaginable. However, as of today, reported prediction accuracies when using plain sequence data in such approaches are typically lower than when using array data [24, 25]. With known pedigrees [26] and/or founder lines with higher read-depth [27] even a lower average read-depth was shown to be useful for genomic prediction, although the predictive ability is still slightly below that of array data. A key limitation of NGS is, that the cost of sequencing increase almost

linearly with the sequencing depth [28]. As a result, generating sequence data with adequate read-depth is still too costly for most routine applications. Thus, genotyping arrays are still considered the gold standard in high-throughput quantitative genetics.

Importantly, due to stochastic aspects of sequencing in sampling from genomic reads, not all variants are called in whole-genome sequencing at very-low to low depth [7, 29]. In the context of a sequenced population, virtually every variant position displays significant amounts of missing calls, leaving these gaps to be filled prior to subsequent applications. This *in silico* procedure is referred to as imputation. Over the years a variety of approaches for imputation have been proposed [30–35]. The interested reader is referred to Das et al. [36] for a detailed review and comparisons between commonly used imputation software. As tools are typically developed for application in human genetics with high genetic diversity and large reference panels, parameter optimization is mandatory for livestock and crop populations [37]. However, as long as somewhat related individuals are considered and parameter settings are chosen adequately, error rates for imputation of array data are usually negligible [37].

One of the key limitations of imputation when working with low read-depth sequence data has been the challenge of phasing reads, causing imputation error rates to increase notably. In contrast to human and livestock genetics, where phasing is a requirement for imputation, fully inbred and homozygous lines are readily produced in maize [8, 38] and other plant species [39]. Inbred lines are frequently used in breeding to, among others, reduce the length of the breeding cycle, increase the genetic variance and safeguard genetic diversity [8, 40–42]. Without the need for phasing, there is high potential in using very-low to low sequencing depth to genotype a large number of lines and apply efficient imputation to obtain maximum data quality at a minimal cost. Specifically, information on read-depth could be used to support imputation. To our knowledge, none of the existing imputation approaches currently addresses this.

In this work, we propose a new imputation pipeline ("HBimpute") for sequence-derived genomic data of homozygous lines that uses long-range haplotype blocks from the software HaploBlocker [43], with haplotype blocks in HaploBlocker indicating cases of group-wise Identity-by-descent (IBD) [44]. This information serves to artificially merge reads of lines in the same haplotype block to locally increase the read-depth, increase calling accuracy and precision, and reduce the proportion of missing calls. The performance of our method is compared to state-of-the-art software. To do so, we will consider BEAGLE 5.0 [35], as the most commonly used software for genomic imputation in plant breeding, STITCH [34], a software specifically designed for the use for low read-depth sequence data, and BEAGLE 4.1 [45], as an example of a software that utilizes genotype likelihoods. Imputation in this manuscript refers to the completion of a dataset with sporadically missing genotypes but not an increase of the marker density by the use of a reference panel. All considered approaches are compared based on the similarity of the imputed dataset with array data and high-read-depth sequence data (30X). Furthermore, the performance of the different imputed datasets is evaluated based on their respective usefulness in a subsequent genome-wide association study (GWAS) and for genomic prediction (GP).

## Results

In the following, we will briefly sketch the key steps of the HBimpute pipeline (Fig 1). As a first step of the pipeline, read-mapping and variant calling are performed to generate a raw SNP-dataset with a potentially high share of missing calls. For this, we suggest the use of FreeBayes [46], but software such as GATK [47] and a workflow along with the GATK best practices [29] is a valid alternative.

Secondly, a haplotype library for the present dataset is derived via the software Haplo-Blocker [43]. This haplotype library is a collection of the identified haplotype blocks in the

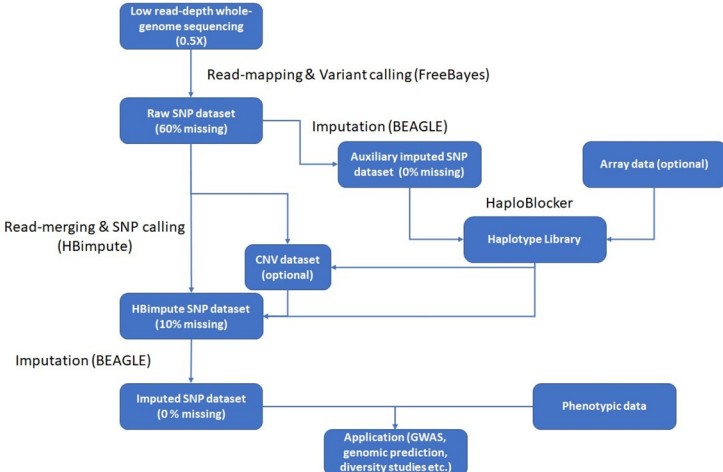

**Fig 1. Schematic overview of the HBimpute pipeline.** The values in brackets indicate the share of missing values in each step for the maize data set with 0.5X sequencing depths.

population, where a haplotype block is defined as a sequence of genetic markers that has a pre-defined minimum frequency in the population and only haplotypes with a similar sequence carry a given haplotype block. Thus, inclusion in the same haplotype block indicates a case of local IBD [43, 44]. As HaploBlocker does not support a high share of missing data, one first has to generate an imputed dataset (auxiliary imputed SNP dataset, Fig 1) and use this set for the calculation of the haplotype library. A potential software to use here is BEAGLE 5.0 [35]. Instead of using the sequence data itself, the haplotype library can also be computed from other genomic data of the considered lines (e.g. array data). In the following, we present results for two alternative approaches, HB-seq and HB-array, depending on whether the haplotype library was derived using the sequence data itself or 600k array data [6], respectively.

Thirdly, the information regarding local IBD from the resulting haplotype library is used in a second variant calling step. In contrast to the initial variant calling, all mapped reads from lines that are locally in the same haplotype block are also used for the respective line. Since the local read-depth in most regions is massively increased via the local merging procedure, an optional step to detect copy number variation (CNV) can be performed. Lastly, the resulting dataset (HBimpute SNP dataset, Fig 1) is imputed via traditional imputing software (imputed SNP dataset, Fig 1) [35] and can be used for subsequent downstream applications.

We applied our imputation pipeline on a dataset of 321 maize doubled haploid lines (DH), derived from an open-pollinated landrace [48]. The DHs were whole-genome sequenced at 0.5X read-depth with 2,152,026 SNPs being called by FreeBayes [46] (compared to 616,201 SNPs on the high-density array [6]). Even though the differences in marker density between the sequence and array data are going down slightly after applying quality control filters, removal of fixed markers, and imputation (1,069,959 vs 404,449 SNPs), this still is a substantial increase in marker density.

When using the HB-seq pipeline, the average read-depth increased from 0.53X to 83.0X. As a result, the share of cells of the matrix containing the genotype data that were called increases from 39.3% before merging to 95.2% after haplotype block merging. Note however that the read-depth varied greatly between lines and genomic regions, as it depends primarily on the frequencies of a given haplotype block in the population. When using HB-array, an average read-depth of 51.3X was obtained with 93.1% of the variants being called. This smaller increase

in average read-depth is mostly due to longer haplotype blocks with fewer lines being identified in HaploBlocker. However, lower read-depth does not necessarily imply lower data quality in HBimpute, as higher read-depth in our pipeline is achieved by merging reads from more and potentially less related lines. In fact, we expect the quality of the array-based haplotype library (HB-array) to be higher than the one obtained via BEAGLE imputed low read-depth sequence data (HB-seq) as the share of missing calls in the raw array data is substantially lower (1.2% vs. 60.7%) [37]. However, in practice, such data is usually not available when sequence data is generated. Note that the reported average read-depth of 83.0X in HB-seq and 51.3X in HB-array does include that reads of the line itself are counted five times to put a higher weighting on the line itself (see Material and methods). Nonetheless, there are still on average 81.0 / 49.3 independently generated reads available for each variant call.

To analyze the performance of our approach, we considered three alternative pipelines for the imputation of the dataset. Firstly, we used BEAGLE 5.0 [35]. Note that the auxiliary imputed SNP dataset exactly corresponds to the finally imputed dataset in BEAGLE 5.0, as the same filtering criteria were used as in our pipeline. Secondly, we used BEAGLE 4.1 [45] because, in contrast to new versions of the software, it is able to utilize genotype likelihoods which have shown to be more accurate for imputation of low read-depth sequence data of non-inbred material [49]. Finally, we used STITCH [34], a method that is designed for use with low read-depth sequencing data. As STITCH is not providing genotype calls for all cells of the dataset, the remaining missing positions were imputed by the use of BEAGLE 5.0. In all applications of BEAGLE 4.1 & 5.0 the effective population size parameter was adapted as this was shown to substantially decrease imputation error rates for datasets with lower diversity than outbred human populations (ne = 10,000; [35, 37]), and STITCH used the comprehensive 'diploid-inbred' mode [34]. Together, these three approaches should represent the current state-of-the-art of methods for the imputation of low read-depth sequence data.

## Imputation

When comparing discordance rates of the imputed SNP dataset with the genotype data from the 600k Affymetrix Axiom Maize Genotyping Array [6], error rates overall are reduced from 0.89% in BEAGLE 5.0 to 0.54% in the HB-seq pipeline and 0.47% in the HB-array pipeline (Table 1). Error rates here refer to the discordance rates between the respective imputed panel and the 600k array data. The dataset was split into three classes to further assess the performance of the imputation (Table 1):

1. Cells first called in FreeBayes step ("Present in raw-data")

2. Cells first called in HBimpute step ("With call after HB")

**Table 1. Discordance rates between the imputed sequence data and the 600k array data depending on the used imputation pipeline.** * For cells with a genotype call in STITCH itself, discordance rates were only 0.39% compared to 0.44 / 0.39% for HB-seq / HB-array for the same entries.

| Pipeline | HB-seq | HB-array | BEAGLE 5.0 | BEAGLE 4.1 | STITCH |
|---|---|---|---|---|---|
| Overall | 0.54% | 0.47% | 0.89% | 3.37% | 1.49%* |
| Present in raw-data | 0.18% | 0.17% | 0.27% | 1.91% | 1.28%* |
| With call after HB | 0.18% | 0.21% | 0.83% | 2.90% | 0.94%* |
| Without call after HB | 7.98% | 5.97% | 11.62% | 20.81% | 9.71%* |
| Imputation accuracy | 0.7610 | 0.7670 | 0.7507 | 0.6653 | 0.6327 |
| REF allele | 0.35% | 0.30% | 0.59% | 1.78% | 0.74%* |
| ALT allele | 0.87% | 0.74% | 1.39% | 6.01% | 2.75%* |

3. Cells first called in the imputed SNP dataset ("Without call after HB")

For all three classes improvements in calling accuracy are obtained with the highest gains for those cells that were first called in the HBimpute step, as the average error rate here is reduced from 0.83% to 0.18 / 0.21% in HB-seq / HB-array. Discordance rates for cells already called in the FreeBayes step are reduced by about 40% as calls are overwritten (0.27% vs. 0.18 / 0.17%, Table 1) when a high number of lines in the same block carry the other variant, indicating the power of our approach to detect calling errors. As the imputed dataset in HB-array was compared to the same array data that was used for the calculation of the haplotype library, we expect results for HB-array to be potentially slightly downward biased. However, as similar improvements were observed when comparing the imputed data panel to high read-depth sequence data this effect should be negligible. Due to the overall higher data quality and lower share of missing markers after the HBimpute step, even error rates for cells imputed in the subsequent BEAGLE 5.0 imputation step are also slightly reduced.

The use of genotype likelihoods in BEAGLE 4.1 led to far inferior results with overall error rates of 3.37%. The STITCH pipeline also led to much higher overall error rates (1.49%). In contrast, those cells of the genotype dataset that were imputed by STITCH itself (and not the downstream imputation with BEAGLE 5.0) were called with very high precision (error rates of 0.39% compared to 0.44 / 0.39% in HB-seq / HB-array). Nonetheless, about 23% of all entries were not called. This is particularly problematic as the minor variants in a high number of markers were not called / identified, resulting in a substantial loss of genetic variation. When analyzing the error rates for a genetic variant depending on the frequency of the variant, we observe that BEAGLE 5.0, HB-seq, and HB-array performed similarly on rare variants, but the two HBimpute-based approaches led to lower error rates for variants with an allele frequency higher than 0.1 (Fig 2). BEAGLE 5.0, HB-seq, and HB-array performed substantially better than BEAGLE 4.1 and the STITCH pipeline for all minor variants (frequency < 0.5). Note that even for the 30X data, discordance rates of 0.30% between the array and sequence data were observed, which can be seen as a lower limit for the achievable error rates of the imputing methods.

When comparing discordance rates of the imputed sequence data to the 30X sequence data that was generated for seven of the considered lines, we again observe much better results in the dataset imputed via our suggested pipeline (HB-seq: 0.98% / HB-array: 0.86%) compared to imputation via BEAGLE 5.0 (1.53%, Table 2). In contrast to the comparison with the array data, error rates for cells filled / called in the HBimpute step are even lower than for markers called in the FreeBayes step, as overwriting of already called variants requires stronger evidence than calling a previously missing variant. Even though overall error rates seem to be higher when compared to the high read-depth sequence data, this is mostly due to lower overall error rates in SNPs that were placed on the array. When just considering marker positions that are also on the array error rates reduce to 0.84% for HB-seq, 0.71% for HB-array, and 1.36% for plain BEAGLE 5.0 imputation [35]. Cells with no called variant in the 30X sequence data were ignored here. Results for BEAGLE 4.1 and STITCH are very similar to the evaluation based on the array data, with STITCH again performing very well on cells that were called by the software itself, but substantially higher overall error rates. Error rates depending on the respective allele frequency are given in S1 Fig.

The results of the imputation accuracy analysis, i.e., the correlation between imputed and real genotypes, yielded very similar results in both comparisons with the highest imputation accuracy in HB array (0.7670 / 0.6698; Tables 1 and 2). Due to the higher relative weighting of the rare variants, the imputation accuracy in STITCH when compared to the array data is lower than in BEAGLE 4.1 (Table 2). When using the array as the true underlying panel, error

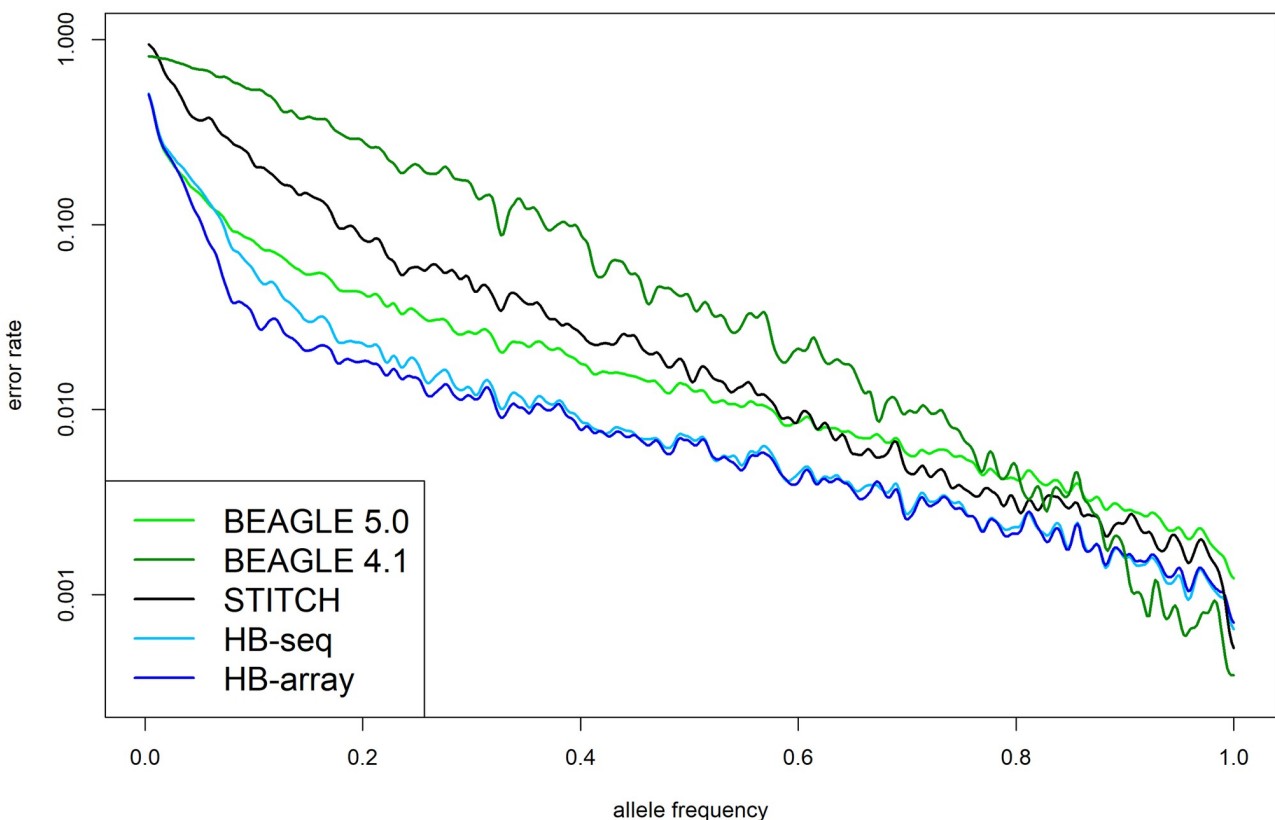

**Fig 2. Discordance rates of the imputed sequence data to the 600k array data depending on the used imputation pipeline and the allele frequency of the given variant.**

rates for the REF allele were half that of the ALT alleles. When using the high read-depth sequence data the opposite was the case with higher error rates for REF alleles. As this should be mainly caused by differences in the allele calling between the array and sequence data and not by imputation, this was not further analyzed in this study.

The final data panels obtained from the sequence data (HB-seq, HB-array, BEAGLE 5.0) contain about three times as many bivariate markers as the array data. The shape of the allele frequency spectrum (S2 Fig) is very similar, indicating a similar increase in the number of available variants in all allele frequencies. When just considering marker positions that are overlapping with the 600k array, a higher share of rare variants (<1%) can be observed in the

**Table 2. Discordance rates between the imputed sequence data and high read-depth sequence data depending on the used imputation pipeline.** * For cells with a genotype call in STITCH itself, discordance rates were only 0.59% compared to 0.78 / 0.68% for HB-seq / HB-array for the same entries.

| Pipeline | HB-seq | HB-array | BEAGLE 5.0 | BEAGLE 4.1 | STITCH |
|---|---|---|---|---|---|
| Overall | 0.98% | 0.86% | 1.53% | 5.05% | 1.93%* |
| Present in raw-data | 0.30% | 0.29% | 0.55% | 2.73% | 1.53%* |
| With call after HB | 0.24% | 0.30% | 0.60% | 3.96% | 1.04%* |
| Without call after HB | 10.63% | 8.43% | 14.46% | 25.80% | 11.39%* |
| Imputation accuracy | 0.6640 | 0.6698 | 0.6528 | 0.5785 | 0.6268 |
| REF allele | 1.70% | 1.47% | 2.31% | 7.32% | 2.53%* |
| ALT allele | 0.60% | 0.54% | 1.13% | 3.88% | 1.62%* |

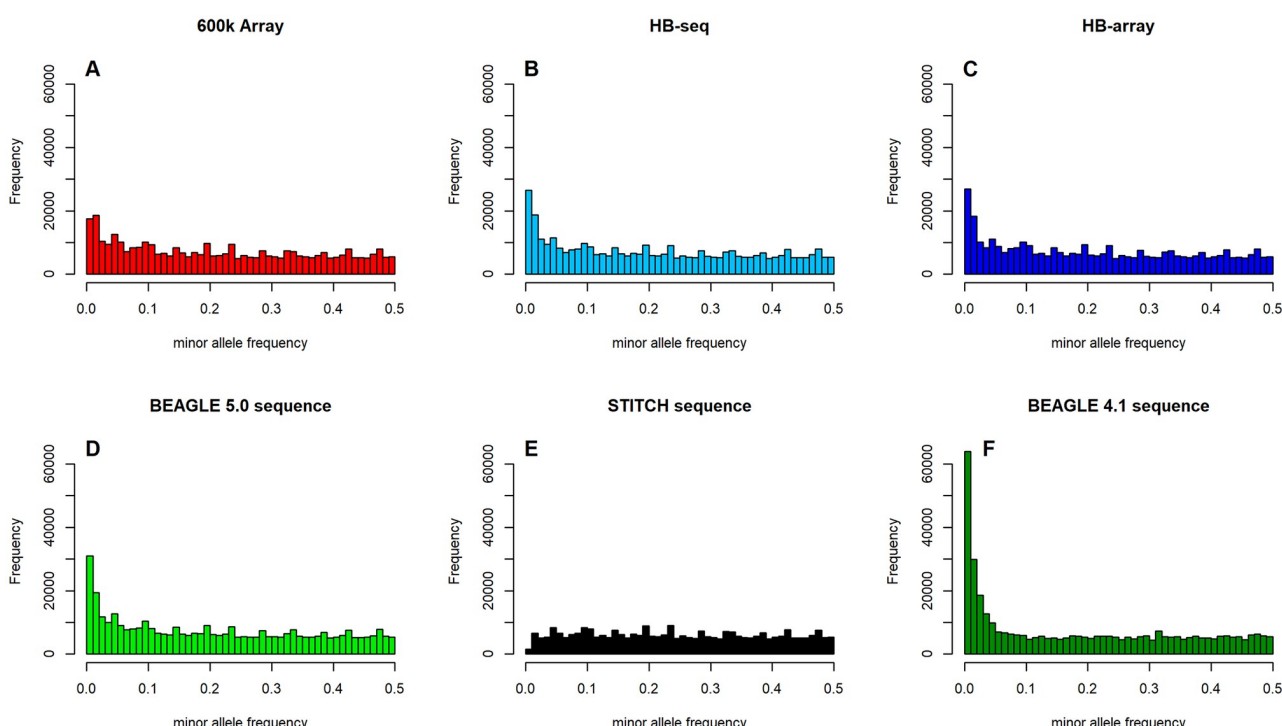

**Fig 3. Allele frequency spectrum of the genomic datasets for all bivarite markers that are shared between the array and sequence data panels.**

sequence data (Fig 3B–3F). As the minor variant is more difficult to impute and the share of called variants before imputation is much higher for the array data (98.8% vs. 39.3%; [37]) this distortion in favor of the more frequent variant should be expected for sequence data. The total number of non-fixed markers that are shared between array and sequence data imputed in HB-seq or HB-array are similar with 366,822, 368,095, and 369,211 SNPs, respectively. In contrast to that, only 299,371 SNPs show variation in STITCH, again showing the tendency of the method to lose minor variants. On the other hand more SNPs (381,728 / 377,900) exhibit variation in BEAGLE 4.1 / 5.0. Additionally, a shift of the allele frequency spectrum towards rare variants can be observed in both BEAGLE methods (Fig 3D and 3F). This shift is caused by markers with medium frequency in the other approaches being more frequently imputed with the major variant and fixed markers exhibiting some variation in BEAGLE 4.1 & 5.0. As in particular variant calls for the rare variants should be more reliable in high read-depth data and array data (as they contain a much lower share of missing calls), we assume that the allele frequency spectra of the 600k data, HB-seq, and HB-array are more reliable for the given marker set.

## Estimation of local read-depth and structural variation

Calling of structural variation from read-mapping typically requires a higher sequencing depth than calling SNPs. When comparing the obtained locally smoothed read-depth of the 30X sequence data to the imputed low sequence data, we observed an average correlation of 0.750 compared to 0.257 for the raw 0.5X data, indicating that the imputed data can be used for the calling of structural variation (correlation without local smoothing: 0.442 vs 0.102). The visual inspection of local read-depth also shows that peaks (Fig 4A and 4C) and local pattern (Fig 4B and 4D) between the low read-depth sequence data imputed via HB-seq and the high read-

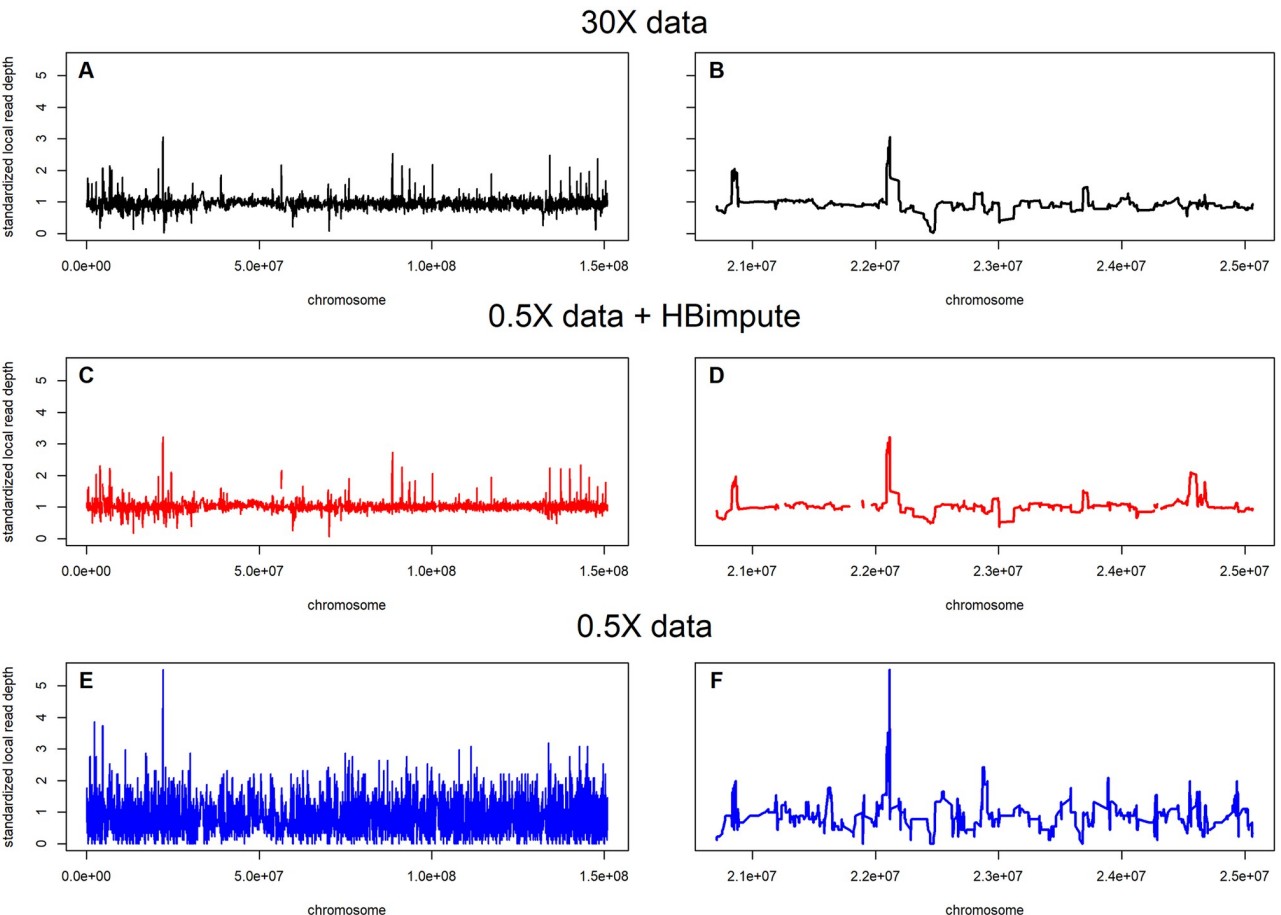

**Fig 4. Estimated standardized read-depth for line PE0213 via the use of high read-depth sequence data (A/B), imputed low read-depth sequence data via HBimpute (C/D) and raw low depth depth sequence data (E/F) for chromosome 10 (A/C/E) and an exemplary chosen segment in a peak region (B/D/F).**

depth sequence data mostly match, whereas the raw low read-depth sequence data has much higher volatility (Fig 4E and 4F). Of the 7,430 markers with a smoothed read-depth above 1.5X in the 30X data, 5,813 (78.2%) were also identified using HB-seq, while only 4,888 (65.7%) were identified in the plain 0.5X data. However, the total number of markers with smoothed read-depth of above 1.5X in HB-seq was only 7,490 (share false-positives: 22.4%) compared to 53,522 (90.9%) in the plain 0.5X data. This suggests a much lower false-positive rate of CNV calls in HB-seq compared to the raw 0.5X data. As HBimpute can only provide an estimated read-depth for regions that are in a local haplotype block, this led to some gaps in the read-depth estimation (4.1%, Fig 4C and 4D).

## Genomic prediction

The performance of the datasets resulting from the different imputing approaches was evaluated regarding their usability for genomic prediction. In addition to the imputed sequence data, we also considered array data from a 600k array and two down-sampled variants to obtain artificial 10k and 50k arrays. For this, we compared the obtained predictive ability of each set for nine traits, including early vigor and plant height at different growing stages, days to silking, days to tassel and root lodging [48]. We define the predictive ability as the

**Table 3. Average predictive ability for nine maize traits [48] depending on the genotype data used for prediction.**
The panel of overlapping markers includes all markers included in the array and sequence data panel after quality control filtering.

| Pipeline | Predictive ability | Predictive ability (overlap) |
|---|---|---|
| 600k array | 0.5170 | 0.5174 |
| HB-seq | 0.5148 | 0.5185 |
| HB-array | 0.5144 | 0.5182 |
| BEAGLE 5.0 | 0.5143 | 0.5177 |
| BEAGLE 4.1 | 0.5099 | 0.5159 |
| STITCH | 0.5136 | 0.5178 |
| 50k array | 0.5143 | 0.5177 |
| 10k array | 0.5159 | 0.5133 |
| HB-seq + CNVs | 0.5126 | 0.5147 |
| HB-array + CNVs | 0.5123 | 0.5143 |

correlation between the estimated breeding values and phenotypes in the test set. The predictive ability for the imputed sequence data panels was marginally lower for eight of the nine considered traits compared to the 600k array. Differences between data panels were however small as the average difference was only 0.22% and at most 0.62% (Table 3 and S1 Table). Remarkably, when using only the marker positions that are shared between the sequence and the array data, minor improvements were obtained for eight of the nine traits (paired t-test, p-values $< 10^{-15}$). As differences on average are just 0.11% this should still be negligible in practice. Nevertheless, it implies that sequence data may, after filtering, have higher precision than array data. Including CNV calls from the HBimpute pipeline led to slightly reduced predictive abilities.

## Genome-wide association study

Furthermore, we evaluated the suitability of the imputed low read-depth sequence data to be used in a GWAS. Our goal was to estimate whether the higher number of variants genotyped compared to the array impacts the power or resolution of GWAS. When comparing the Manhattan plots derived based on sequence data and array data on simulated traits, in general, higher peaks are observed for all panels with sequence data, leading to a higher number of regions identified when using the same p-values. To correct for this, we instead report the share of true positive QTL hits compared to the total number of regions with a GWAS hit. This results in a line of potential outcomes depending on the used significance threshold (Fig 5A and S2 Table). Thus, a realization with a higher number of identified real QTLs combined with a higher share of true positives can be seen as a strict improvement of the results (Fig 5A). Overall, results between the sequence data panels imputed via HB-seq, HB-array, BEAGLE 5.0, and STITCH yielded very similar results and were all slightly better than the results when using the 600k array data. Between the different imputing approaches, HB-array performed best when low significance thresholds are used (and thus more identified real QTLs), while STITCH performed best with a high significance threshold. However, the differences between data panels are only minor. In addition, differences are not only impacted by the imputation but also by the differences in the initial variant calling (FreeBayes, direct ascertainment from the 600k array, STITCH). For all sequence-based data panels and in particular the FreeBayes-based datasets (HB-array, HB-seq, BEAGLE 5.0), some isolated GWAS hits were observed. Thus, resulting in separate identified QTL regions that were then classified as false positives. A potential reason for this could be transposable elements and other types of structural variation

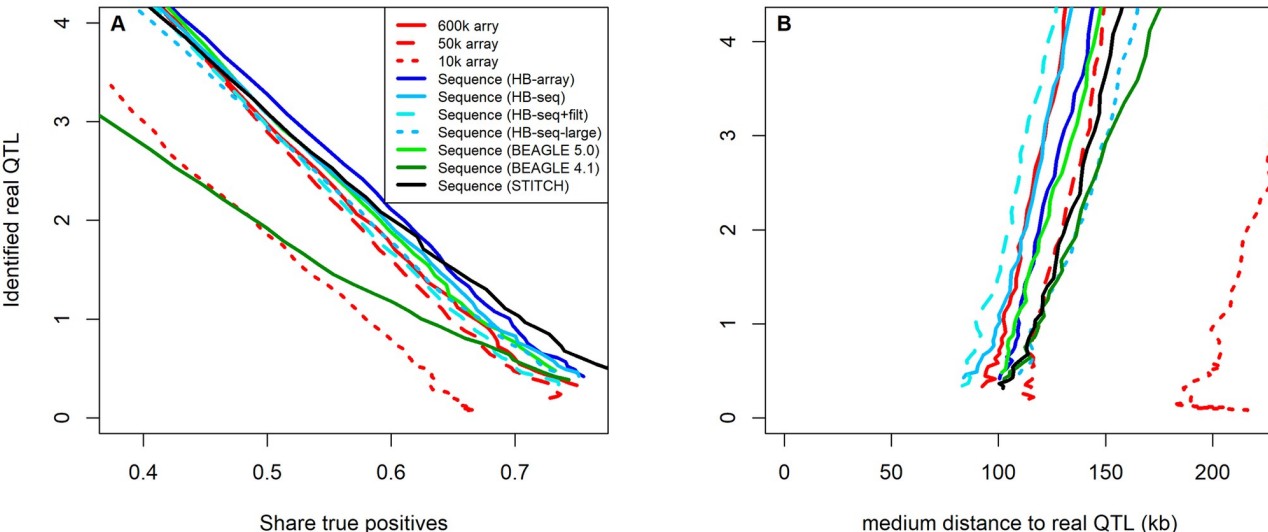

**Fig 5.** Number of positive GWAS hits for simulated traits with 10 underlying QTL depending on the share of true positive hits (A). Median distance of the local GWAS peak (highest p-value) and the underlying true QTL for correct GWAS hits (B).

as the B73v4 reference genome [50] represents dent germplasm whereas the lines in this study belong to the flint gene pool [51].

Results for the 600k array were slightly better than for the 50k array and substantially better than for the 10k array. In contrast to genomic prediction, results when applying stronger filtering by only using the markers also present on the array (HB-seq+filt) led to slightly worse results than HB-seq. On the contrary, increasing the number of considered markers by the use of weaker filtering criteria did not improve results. This was the case for both weaker filtering in the HBimpute step (1.4 million SNPs; HB-seq-large) and weaker filtering in the initial variant calling in FreeBayes [46] or GATK [47] (not shown). As linkage disequilibrium in the considered dataset of a European maize landrace is high [43], we would in general not expect much information gain for these datasets in the first place.

In terms of mapping power, we observed the lowest median distance between the GWAS peak (highest local p-value) and the underlying true QTL when using HB-seq data when only including markers shared with the array (Fig 5B), closely followed by the 600k array data and the sequence data imputed via HB-seq, HB-array or BEAGLE 5.0. Indicating that for fine-mapping, marker quality should be more important than the total number of markers. Worst results were obtained for the down-sampled array with 10k markers, indicating a substantial information loss caused by the lower marker density.

## Discussion

HBimpute is shown to be a pipeline for accurate imputation of low read-depth sequence data. Results indicate that the use of HBimpute allows sequencing at reduced read-depth while maintaining high data quality that is comparable to high-density array data. Thus, HBimpute leverages significant cost savings and / or higher data quality for subsequent applications.

When comparing the different imputation approaches, the use of the genotype likelihood in BEAGLE 4.1 was not beneficial, as the genotype likelihood in our particular case of doubled haploid lines provides relatively limited additional information. In addition, BEAGLE 4.1 is not designed for use with fully homozygous data, which here seems to have a higher impact than the information gain.

Observed error rates for the STITCH pipeline (including subsequent BEAGLE 5.0 imputation) were much higher than for HBimpute, while the variants called in the STITCH step itself were actually competitive with HBimpute (77% of all calls). In principle, one could even consider the use of STITCH to derive the auxiliary imputed SNP dataset in HBimpute. When first computing the imputed SNP dataset based on HB-seq, then replacing all cells with a call in the STITCH step and using this dataset as the auxiliary imputed SNP dataset in a second run of the HBimpute pipeline, a further absolute reduction of the error rate by about 0.03% was obtained (not shown). As the overall complexity of the pipeline is substantially increased, separate Variant Call Format (VCF) and Binary Alignment Map (BAM)-files need to be processed and the overall computational load is substantially increased, this will only be of practical relevance in very specific cases.

Overall, we conclude that WGS and GBS are valid alternatives to genotyping arrays for the generation of genomic data and use in subsequent applications. In particular for genomic prediction, the use of HBimpute improved results slightly compared to the other state-of-the-art methods for the imputation of low read-depth sequence data. Results for the sequence data were even slightly better than those for the array data when using the same set of markers overlapping between sequence and array data to avoid that difference caused by the use of a better-suited marker panel. Importantly, this may indicate that the overall data quality of low read-depth sequence data is higher or at least on par with high-density array data. When using a larger set of SNPs for the sequence data, our results are in line with other studies that suggest slightly lower predictive ability when using sequence data [24, 25, 52]. As a consequence, we conclude that the overall quality of markers that are not on the array is lower. As array markers are typically ascertained based on quality and positioned in conserved regions, this is expected. In particular, it does not mean that the data quality for the same variants in low read-depth sequence data is actually lower. However, in agreement with Erbe et al. [53], even the use of a 10k array led to basically the same prediction accuracies and could therefore be a cost-competitive genotyping alternative when the only intended subsequent application is genomic prediction and such an array exist for the respective species.

In the GWAS study, HBimpute yielded slightly better results than those obtained with the use of sequence data imputed via STITCH or 600k array data, while substantially outperforming sequence data imputed with BEAGLE 4.1. In particular, in terms of overall ability to detect QTLs, the sequence data panels (HB-seq, HB-array, STITCH) outperformed the 600k array data. In terms of fine-mapping, both HB-seq and HB-array were at least on par with the 600k array data and better than STITCH. In contrast to genomic prediction, a much higher effect of the marker density was observed, with reduced panels for both the array data and sequence data yielding substantially worse results. This is in line with other studies that observed better GWAS results with increasing marker density [13, 54]. A further increase of the marker density by weaker quality filtering did not further improve results. The results regarding GWAS should still be taken with a grain of salt, as all considered traits were simulated with effect markers being partially based on the 600k and partially based on the HB-seq data. Thus, results should be slightly biased towards these methods. However, as the HB-seq data still performed better than STITCH for the 600k-based QTLs, results should still be robust in the overall context.

The inclusion of CNVs did not yield better performance in genomic prediction or GWAS. As the overall quality of CNV calls should be lower than marker calls, this is most likely due to the overall lower data quality. By design, these CNV calls actually only introduced noise to the GWAS, as no effects were placed on CNV calls in the simulation. As real traits were used for genomic prediction this is not the case here. Nonetheless, CNV calls can still be of interest when analyzing specific regions of the genome as a follow-up of an initial GWAS analysis.

Further, we would still assume that there are some high-quality variants in both the CNV panel and the panel of non-array markers. Identifying these high-quality variants and applying better filtering strategies than just using exactly the set of markers overlapping with the array could potentially be a way to further improve results in downstream applications.

When just considering genotype information on the panel of overlapping markers between sequence and array data the predictive ability was marginally improved, indicating that the overall data quality of low read-depth sequence data is on par or even slightly higher than array data. This is further supported by higher imputing error rates on non-array markers and slightly increased predictive ability when using HBimpute instead of BEAGLE 5.0 for imputation of the sequence data.

Overall, we can conclude that rating the usefulness of a genomic dataset is highly dependent on the intended downstream application and data preparation, and filtering should be chosen accordingly. With increasing marker density in sequence data, calling and imputing errors will increase (due to the inclusion of low-quality markers) and an adequate weighting between marker density and quality has to be found. For example, when conducting a GWAS focus should be on including a high number of markers, whereas for genomic prediction high-quality markers have shown to be more important. Here, one could even consider further cost savings by the use of smaller genotyping arrays [53]. In this context, HBimpute is providing a framework to improve imputation accuracy and thereby improve data quality compared to existing imputation software. Generally, both GWAS and genomic prediction via a mixed model are quite robust methods that will neutralize most of the issues associated with partially poor data quality.

The use of sequence data comes with both challenges and opportunities. Sequence data provides more information in less conserved regions and hence provides more information on structural variation of the genome [55]. In particular, several crop genomes have a high share of transposable elements (e.g. 85% in maize [56]). Marker data in those regions is typically noisier than array markers that are specifically selected to be in more conserved regions [6, 14]. Note that high-quality genotyping arrays are not available for all species and the relative cost of sequencing will be lower for species with short genomes. Therefore, the decision on which genotyping technology to use in practice will be highly dependent on the species at hand, its genome length, available genotyping arrays, and intended subsequent applications.

A key limitation of the HBimpute pipeline is that it requires highly accurate phase information that is typically not available for low read-depth sequence data in non-inbred material and therefore is mainly applicable to inbred lines. However, with the availability of long-read sequencing technologies and highly related individuals with available pedigree information, as commonly present in livestock genetics, this might change in the near future. The here proposed HBimpute pipeline and software can be applied on heterozygous data in the same way as with inbreds by handling the two haplotypes of each individual separately.

In particular, for the detection of CNVs, the here suggested pipeline is shown to be highly efficient, as the estimated local read-depth of the imputed 0.5X data was very similar to 30X data that was generated for seven of the studied lines. At this stage, this can be seen as a first proof of concept that shows the potential of our approach. Nevertheless, the overall data structure obtained via HBimpute is substantially different from raw sequencing data, despite a large increase in the artificial read-depth in the dataset. Crucially, the local read-depth does not just depend on the sequencing depth, but the number of lines in a local haplotype block. Thus, existing methods for calling of CNVs and structural variation, in general, can not be applied straightforwardly, but rather the development of new approaches is required. Calls for structural variation for different lines within the same local haplotype block will usually be very similar. Thus, parameter adaption in HaploBlocker can be used to adapt the structure of the used

haplotype library. Thus, one can control how similar lines in the same haplotype have to be to put a focus on population-wide or within-population differences. Still, as other studies detecting structural variation typically rely on at least 5X sequence data [22, 23], our approach could enable a large cost reduction and the calling of structural variation in large-scale populations.

## Materials and methods

In the following, we will describe the haplotype block-based imputation step of our proposed pipeline in more detail. This step is applied after an initial SNP calling step that is resulting in a dataset, we refer to as the raw SNP dataset (Fig 1). In our study, each of the 340 individual DH lines had its raw read file (FASTQ) aligned to the B73v4 reference genome [50] using BWA MEM [57]. Subsequently, variant calling in FreeBayes was performed using 100 kilo-base pair genome chunks with marker positions from the 600k Affymetrix Axiom Maize Genotyping Array [6] given as input to force variant reporting at those locations (-). Furthermore, 5 supporting observations were required to be considered as a variant (-C 5) with at most 3 alleles per position (–use-best-n-alleles 3) and a maximum total depth in a position of 340 (–max-coverage 340). To ensure adequate data quality, markers with more than 1% heterozygous calls were removed since we would not expect heterozygous genotypes for DH lines. Subsequently, 19 lines were removed from the panel, as genomic data from the 600k array and sequence data showed strong indication for sample contamination and / or mislabeling (see Genotype data used subsection).

The newly proposed HBimpute step is using the raw SNP dataset (Fig 1) as the only mandatory input and can be separated into three sub-steps, that will be discussed in the following subsections:

1. Derivation of a haplotype library

2. Read-merging

3. SNP-calling

Note, that only the reads that are included in the VCF file are used in our pipeline and, in particular, there is no need to access the original raw data from the BAM files or similar in any step of the proposed pipeline. After executing these steps, the resulting HBimpute SNP dataset (Fig 1) is obtained, with only a few remaining missing calls. Nonetheless, subsequent imputation via traditional imputation software is necessary for most downstream applications. In our tests, the software BEAGLE 5.0 performed well both in terms of computing time and accuracy [35] and was chosen for all reported tests. We will here focus on describing the default settings of the associated R-package HBimpute, but also discuss potential deviations with most parameters in the pipeline being adaptable to set a weighting between imputation quality, the number of markers considered, and the overall share of markers called in HBimpute.

Individual steps of the procedure will be explained along the example dataset shown in Fig 6 with five haplotypes and ten markers each. For simplicity, we are assuming a read-depth of one for all called genotype entries.

### Derivation of the haplotype library

In the first step of the HBimpute, the objective is to derive a haplotype library via the associated software HaploBlocker [43]. As HaploBlocker itself is not supporting a high share of missing data, the raw SNP dataset first needs to be imputed to generate an auxiliary imputed SNP dataset (Fig 1). Alternatively, other genetic data of the considered lines like array data can also be used. Results for both approaches (HB-seq & HB-array) are presented in the Results section.

Raw SNP dataset

HBimpute SNP dataset

**Fig 6. Toy example for the HBimpute step.** Each column represents a SNP and each row represents a haplotype (for inbred lines: individual). Haplotype blocks are indicated by colored blocks. The blue and red block are overlapping.

Since the overall data quality in terms of consistency and overall calling precision in the array data should be higher than the raw low read-depth sequence data, the use of array data is recommended when available (HB-array). Furthermore, additional lines can be included as a reference panel in both approaches. Individuals in the reference panel can either be used to improve the quality of the haplotype library and / or provide additional reads to be used in the subsequent read-merging step. In all our tests, the parameter settings in HaploBlocker were adjusted to identify long haplotype blocks which are potentially present in low frequency (node_min = 3, edge_min = 3, weighting_length = 2 [43]) and a target coverage was set to ensure sufficient coverage of the haplotype library (target_coverage = 0.95 [43]). For datasets with less relatedness between lines, a reduction of the window size might be needed to detect shorter haplotype blocks. This is only recommended when the expected length of haplotype blocks is similar to the window size in HaploBlocker (default: window_size = 20). For reference, haplotype blocks in both HB-seq and HB-array blocks had an average length of more than 1'000 SNPs. Alternatively, one can also consider using an adaptive window size (adaptive_mode = TRUE [43]). As this comes with a substantially increased computing time and should not affect results when haplotype blocks are substantially larger than the window size in HaploBlocker, this is usually not needed.

For our toy example given in Fig 6, three blocks are identified with the red block including haplotypes 1,2,3 spanning over SNPs 1–10, the green block including haplotypes 4,5 spanning over SNPs 1–5, and the blue block including haplotypes 1,2,3,4 spanning over SNPs 6–10.

## Read-merging

The output of HaploBlocker is a haplotype library. As the contained haplotype blocks indicate cases of group-wise IBD [44] this means that all included haplotypes should have locally matching sequences and that all reads of these lines can be used for the subsequent SNP-calling. In case a line is part of multiple haplotype blocks, reads of all lines in either of the two haplotype blocks are used. To still be able to detect recent and rare variation, the reads of the line itself are used with a higher weighting in subsequent steps (default: five times as high). Variant calls that are missing in the initial variant calling in FreeBayes [46] and are only imputed in the step of the derivation of the haplotype library are ignored in this step. In our example, this means that for marker 1 in haplotype 1 there are no reads supporting variant 0 and two reads supporting variant 1. Similarly, for marker 5 there are five reads supporting variant 1 and only one read supporting variant 0 as the read of the haplotype itself is counted with a higher weighting. In a haplotype library from a real genomic dataset, each block usually contains far more haplotypes and therefore a much lower relative weighting is put on the haplotype itself.

## SNP-calling

After the read-merging step, a further SNP calling step is necessary. Since it is neither possible nor necessary to obtain calls for all markers in this step, the focus here is on retrieving calls for markers with clear evidence of a certain variant. In our case, this means that at least 80% of all reads are supporting the same variant. In case no call was obtained in this step, but a variant was called in the original raw SNP dataset, this variant is inserted. This is mainly done to avoid losing rare variants.

In the toy example (Fig 6), in marker 5 variant 1 is called for haplotype 1 as five of the six reads considered support variant 1. Even though haplotype 2 is in the same local haplotype block variant 0 is called here, as the reads of the line itself are weighted higher. For haplotype 3 no variant can be called as both variants are supported by exactly one read, thus not exceeding the 80% threshold.

**Quality filters.**   All markers with an estimated read-depth that is below 50% of the overall mean read-depth are removed from the dataset to ensure data quality. Similarly, all markers with more than 50% missing calls are removed. These settings can be seen as relatively conservative as only markers with extremely low call rates are removed. Thus, the introduction of potential noise from low-quality markers in the subsequent BEAGLE 5.0 imputation procedure is reduced. Further increasing filter thresholds will increase calling precision but also potentially result in the loss of usable information.

**Optional: CNV-calling.**   As the read-depth after the HBimpute-based SNP-merging is massively increased, the SNP-calling step can be combined with an optional step to detect CNVs. To negate issues of high per-marker variance in read-depth, we first apply a kernel smoothing function to estimate the local read-depth of the population. This is done via a Nadaraya-Watson-estimator [58] with a Gaussian kernel and set bandwidth (default: 0.25 mega base pairs (Mb)). The local read-depth of a single haplotype is then compared to the population average with regions above 1.3 of the expectation being classified as CNVs and regions below 0.7 being classified as deletions. By adjusting the bandwidth of the smoothing function the resolution of the identification can be adapted to specifically target short / long CNV segments. This approach will not detect other types of structural variation such as translocations, inversions, or insertions as not all raw reads from the BAM file, but only aligned reads that were used for the variant calling in the VCF-file are used here. Instead of performing the HBimpute step on the VCF-file, merging could also be directly applied to the reads themselves, followed by a second run of a variant caller.

For simplicity reasons in the toy example (Fig 6), we are assuming here that only the marker itself is impacting the CNV calling in a given marker and thus no local smoothing is applied. The average read-depth in marker 4 is 0.4X as two of the five included haplotypes were called. Haplotypes 4,5 have an estimated read-depth of 0 as no variant was called. Haplotype 1 has an estimated read-depth of 0.285X (two reads for seven haplotypes) as the haplotype itself is counted five times. Both Haplotype 2 and 3 have an estimated read-depth of 0.857X (six reads for seven haplotypes). This would lead to deletions being called for haplotypes 4 and 5 (0X / 0.4X < 0.7) and duplications being called for haplotypes 2 and 3 (0.857X / 0.4X > 1.3). This small-scale toy example is not constructed for the identification of CNVs and a much higher number of supporting reads and local smoothing is usually required for the detection of copy number variation. Both deletions and duplications are thereafter added as an additional binary marker that is coding if the respective structural variation is present or not.

Other basic single SNP or window-based approaches on the read-depth were also tested [59], but had limited success. No testing has been done with split read or assembly approaches [60] as all analyses in HBimpute used the VCF-file as input. Methods should however be

relatively easily extendable to such approaches to enable the detection of other types of structural variation.

## Heterozygous data

In principle, the same pipeline suggested for inbreds can also be applied on diploid / heterozygous data that is using the two respective haplotypes separately. However, as the phasing accuracy of low read-depth sequence data is usually low, the derivation of an accurate haplotype library is heavily impacted by the software used for the initial phasing, leading to results of the SNP-calling being very similar to the original phased and imputed datasets from the respective external software (not shown). With advances in long-read sequencing [61], the phasing quality might improve in the future.

## Genomic prediction

The usability of the different datasets for genomic prediction was evaluated by comparing each set for its predictive ability for nine real traits, including early vigor and plant height at different growing stages, days to silking, days to tassel, and root lodging. The dataset was split into 280 lines used for model training and 41 lines as the test set and evaluation of the performance was done based on the average predictive ability. We define the predictive ability as the correlation between the estimated breeding values and the phenotypes in the test set. For the evaluation a linear mixed model [62] with a genomic relationship matrix [63] was used (genomic best linear unbiased prediction). This procedure was repeated 1,000 times for all considered traits.

## Genome-wide association study

To compare the performance of the imputed datasets, a genome-wide association study on simulated phenotypes, and therefore known underlying regions, was conducted. For each trait 10 underlying QTL were simulated with 5 QTL positions randomly drawn and evaluated based on the 600k data and 5 QTL positions drawn and evaluated based on the HB-seq data. The heritability $h^2$ of the simulated traits was assumed to be 0.5, with all 10 QTLs having equal effect size. All GWAS hits, meaning markers below a certain p-value, were put in a joined region in case they were at most 1 Mb apart from each other and a region was considered a positive hit in case the underlying QTL was at most 1 Mb away from the region. The given procedure was repeated for 10,000 separately simulated traits and the GWAS was performed using the R-package statgenGWAS [64, 65]. Applying a minor allele frequency filter is common in GWAS analysis. However, to avoid potential biases caused by differences in the allele frequency spectra (cf. Fig 3) we did not apply any filtering in this study. This should not be a concern as QTLs were only assigned to SNPs with a minor allele frequency of 0.1 or more.

## Genotype data used

For all tests performed in this study low read-depth sequencing data with a target read-depth 0.5X was generated for 340 maize doubled haploid lines, derived from an open-pollinated landrace (Petkuser Ferdinand Rot; [48]). Variants were called using the software FreeBayes [46] with marker positions of the 600k Affymetrix Axiom Maize Genotyping Array [6] being forced to be called. This resulted in a data panel of 2,152,026 SNPs and an average read-depth of 0.73X. 19 lines were removed from the panel as genotype calls between the called variants and independently generated data from the 600k array [48] differed by more than 0.75% indicating sample contamination. Furthermore, re-labeling of 4 lines was performed as genotypes

were matching with different lines based on the 600k array data. As we would not expect heterozygous calls in DH lines all markers with more than 1% heterozygous calls were removed from the panel (34% of all markers). Furthermore, fixed marker positions were also excluded (10% of all variants). Leading to a raw SNP dataset (Fig 1) containing 1,109,642 SNPs (compared to 404,449 variable SNPs with adequate quality (PolyHighResolution [66]) on the high-density array [6] (total: 616,201 SNPs)) with the average read-depth being reduced to 0.53X. After the quality filter in the HBimpute step 1,069,959 SNPs remain. Quality control and imputation for the 600k array were performed as described in Pook et al. [43]. As only 1.2% of all markers were imputed this should have a negligible impact on this study.

## Software

The read-merging and SNP-calling procedure presented in this manuscript are implemented in the R-package HBimpute (available at https://github.com/tpook92/HBimpute). Computing times of the HBimpute pipeline are higher than regular imputation procedures like BEAGLE [35], as the BEAGLE algorithm itself is executed twice and HaploBlocker [43] needs to be applied on the auxiliary imputed SNP dataset (Fig 1). Our pipeline from the raw SNP dataset to the final imputed SNP dataset for chromosome 1 took 107 minutes with 68 minutes spent in BEAGLE 5.0 for the HB-array pipeline. The HB-seq pipeline took 226 minutes as the haplotype library contained significantly more haplotype blocks that had to be processed in HBimpute. For our dataset, peak memory usage in the HB-array pipeline was occurring when performing imputation via BEAGLE 5.0 (4.6 GB of memory). For HB-seq, peak memory was reached in the HaploBlocker step with 15.5 GB of memory. Scaling will be somewhat dependent on the dataset and was approximately linear in both the number of SNPs and individuals for the dataset considered. For datasets with high genetic diversity, the scaling can increase up to a quadratic increase in the number of individuals. For more information on this, we refer to Pook et al. [43]. For reference, BEAGLE 5.0 needed 34 minutes, BEAGLE 4.1 took 100 minutes and STITCH took 21 minutes on the same dataset with a peak memory usage of 2.3, 4.8, 1.4 GB, respectively. All computing times reported were obtained when using a single core in HBimpute on an Intel(R) Xeon(R) E7–4850 2.00GHz processor. Note that these computing times are typically negligible compared to the time needed for preprocessing and the initial variant calling. Thus, higher computing times should not be a major concern here.

The R-package can be directly be installed within an R session via the following command:

```
install . packages("devtools")
devtools :: install_github("tpook92/HBimpute", subdir = "pkg")
```

This pipeline is using the software BEAGLE 5.0 as the backend imputation tool (https://faculty.washington.edu/browning/beagle/beagle.html) [35].

## Supporting information

**S1 Table. Predictive ability for the nine maize traits depending on the genotype data used. Details on the individual traits and growing stages (v3-final) can be found in Hölker et al. [48].**
(DOCX)

**S2 Table. Number of true underlying QTLs identified depending on the false discovery rate (FDR).**
(DOCX)

**S1 Fig. Error rates depending on the allele frequency of the given variant depending on the used imputation pipeline when comparing to 30X sequencing data.**
(TIF)

**S2 Fig. Allele frequency spectrum of the different genomic datasets.**
(TIF)

## Author Contributions

**Conceptualization:** Torsten Pook.

**Data curation:** Adnane Nemri, Eric Gerardo Gonzalez Segovia, Daniel Valle Torres.

**Formal analysis:** Torsten Pook, Daniel Valle Torres.

**Funding acquisition:** Henner Simianer, Chris-Carolin Schoen.

**Investigation:** Torsten Pook.

**Methodology:** Torsten Pook, Adnane Nemri.

**Software:** Torsten Pook.

**Supervision:** Henner Simianer, Chris-Carolin Schoen.

**Validation:** Daniel Valle Torres.

**Visualization:** Torsten Pook.

**Writing – original draft:** Torsten Pook.

**Writing – review & editing:** Torsten Pook, Adnane Nemri, Eric Gerardo Gonzalez Segovia, Daniel Valle Torres, Henner Simianer, Chris-Carolin Schoen.

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
