## [Decision Letter · Decision Letter 0]

17 Mar 2021

Dear Dr Pook,

Thank you very much for submitting your Research Article entitled 'Increasing calling accuracy, coverage, and read depth in sequence data by the use of haplotype blocks' to PLOS Genetics.

The manuscript was fully evaluated at the editorial level and by independent peer reviewers. The reviewers appreciated the attention to an important problem, but raised some substantial concerns about the current manuscript. Based on the reviews, we will not be able to accept this version of the manuscript, but we would be willing to review a much-revised version. We cannot, of course, promise publication at that time.

Should you decide to revise the manuscript for further consideration here, your revisions should address all of the specific points made by each reviewer. In particular, you should more thoroughly compare to both BEAGLEv4.1 and STITCH, where some advantage over these existing approaches should be demonstrated. We will also require a detailed list of your responses to the review comments and a description of the changes you have made in the manuscript.

If you decide to revise the manuscript for further consideration at PLOS Genetics, please aim to resubmit within the next 60 days, unless it will take extra time to address the concerns of the reviewers, in which case we would appreciate an expected resubmission date by email to plosgenetics@plos.org.

[LINK]

We are sorry that we cannot be more positive about your manuscript at this stage. Please do not hesitate to contact us if you have any concerns or questions.

Yours sincerely,

Jonathan Marchini

Associate Editor

PLOS Genetics

David Balding

Section Editor: Methods

PLOS Genetics

Reviewer's Responses to Questions

**Comments to the Authors:**

Reviewer #1: The authors present a method called HBImpute for estimating genotypes from low coverage (e.g. 0.5x) sequence data. The method is designed for the special case of samples with homozygous genotypes (double haploid lines) that arise in plant breeding. The method identifies haplotype blocks and clusters sequence reads from identical by descent haplotypes in each block. Sequence reads for a sample are augmented with sequence reads from other identical by descent haplotypes. The method gives an approximate 40-50% reduction in genotype error rates over a competing method (Beagle v5.0) on an evaluation data set with 0.5x sequence coverage when compared to array genotypes. The HBImpute genotypes were used for association analysis and phenotype prediction, and yielded results that were similar to results obtained for SNP array data. The augmented sequence coverage appears to improve detection and calling of copy number variants. Software implementing the method is freely available for non-commercial use.

Comments

The type of genotype imputation should be clearly delineated in the introduction to avoid potential confusion. It appears that you are imputing sporadic missing genotypes, and not imputing missing markers using an external reference panel, or genotypes from genotype likelihoods. Is this correct. Would methods for imputing genotypes from genotype likelihoods be a better solution here?

Software license restrictions should be noted in the paper.

In the initial VCF file that is produced from low coverage sequence data, what thresholds determine whether a genotype is called or set the genotype to missing?

The term “cells” is used several times on p 4/17. Please define “cell” or use a different term.

In Table 1, what is the input data to Beagle for the results in the “Beagle” column? Is it the imputed data set that is used to generate the haplotype library?

P. 5/17, “observe an increased number of markers for all MAFs”. Increased compared to what?

Figure 4 – What is varying to produce the each curve – presumably it is the significance threshold. This should be stated in the figure text. Also the left figure would be a bit easier to interpret if the order of lines in the legend was consistent with the order the lines in the left figure.

In Methods, can you indicate how overlapping blocks (as in Figure 5) are handled?

P. 6/17, how is “predictive ability” defined?

Figure 5, please describe what the left and right sides represent in the figure text

Line 358. Does the read coverage computed after running HB include up-weighted reads from individuals?

“Markers above a certain p-value” (line 451). Do you mean “above” or “below”?

Reviewer #2: Pook et al. presents a new pipeline, HBimpute, for imputation of low-coverage WGS data, designed for plant genetics. The key feature of the pipeline is to locally merge reads of different lines when they share a haplotype block, to increase the read depth of the genotype calling procedure, leading to better imputed genotypes compared to BEAGLEv5.

The pipeline presented in the manuscript improves the discordance rates of imputed sequence data when benchmarked against SNP array data and high coverage WGS, at a cost of an increased computational time.

I have few questions and comments regarding the manuscript and the benchmarking, which in some places seems to be lacking. In particular, the manuscript seems to focus on benchmarking the HBimpute pipeline against standard imputation (based on hard calls). However, due to the nature of low-coverage WGS, typically this type of data is imputed from genotype likelihoods, rather than hard calls. An evaluation of the HBimpute pipeline and methods specifically based for imputation of low-coverage WGS (BEAGLEv4.1 and STITCH) is therefore needed.

Additionally, other metrics to assess the quality of the genotype calls need to be provided.

Major comments

1. The pipeline uses BEAGLEv5 to firstly impute missing data to derive a haplotype library with HaploBlocker. This is subsequently used to improve the quality of the raw SNP dataset, prior to a final imputation with BEAGLEv5. While I do understand the benefit of this, compared to imputation on the raw SNP dataset alone, I am not convinced that this is the best procedure.

Methods like BEAGLEv4.1 are able to use genotype likelihoods (obtained in your case with Freebayes) instead of hard-called genotypes to produce reference-aware genotype calls, that use information from all other target samples. As BEAGLEv4.1 outputs only positions having at least one read covered, missing likelihoods can be called as uniform, or standard imputation with BEAGLEv5 can be run afterwards, as performed in (Homburger et al., 2019).

The authors should show how their pipeline compares to genotype likelihoods with BEAGLEv4.1 (with uniform likelihoods to replace missing likelihoods) and BEAGLEv4.1+BEAGLEv5.0.

2. As the authors work with low-coverage data where a small number of founders is known or can be derived, they should compare the imputation performance of their pipeline and the method STITCH (Davies et al, 2016), as the method seems to be well designed for exactly the same task.

3. The authors show discordance rates as the only metric for genotype accuracy. They should show how their pipeline performs for different type of variantion (e.g. SNPs vs indels), add both marker-level accuracy (such as Pearson correlation), and additional genotype-level accuracy, to show the difference between REF and ALT calls. Stratifying the accuracy by minor or non-reference allele frequency could also add value to the result section and to distinguish the different methods.

4. All the figures should be drastically improved as often they are hard to read, lack of titles and axes descriptions.

Minor comments

-- Would be interesting to know the impact of the local merging of reads on the reference bias. A quantification of this would add value to the manuscript.

--Page 6, line 170. (connected to point 3) The authors talk about increased power to call structural variation. This statement should be backed up showing the improvement obtained, by using high coverage data as a validation.

--Page 6, line 180, Figure 3B/D. Assuming that Fig B is the high coverage (not very clear), the authors could also explain why there is a peak of read depth in the tail of Fig D.

--Page 6, line 192. (connected to point 3) The authors should validate the CNV calls you derived from the HBimpute pipeline using the 30x data to explain the results of Table 2.

-- Page 7, line 196: This section is hard to follow and might be restructured. I am not sure what are and how strong are the conclusions of this GWAS.

-- Page 7, line 206: What filters have been used prior to the GWAS?

-- Page 7, line 206: authors should elaborate more on the fact that plain BEAGLE imputation gets better results in the GWAS setting than HB-seq, even though it does have bigger discordance rates

-- Page 9, line 301: with 0.5x coverage, it it relatively unlikely to get 5 observations. The authors might want to decrease the threshold, and check if the performance of BEAGLE5 increases.

-- Page 9, line 339: Parameters and reference panels used for BEAGLE are not clear to me.

--Page 11, line 407. There might be a typo regarding the read depth of marker 4

-- Page 11, line 424: Not sure I agree with the statement: “Phasing accuracy of low read-depth sequence data is usually relatively low”. Intuitively, I would think that phasing accuracy (for >0.5x data) decreases increasing the coverage, as the number rare variants to be phased also increases, and these are very hard to phase. An explanation would be useful.

--Page 12, line 480. Computational resources used to run BEAGLE (running time/memory usage) is missing.

Reviewer #3: In this manuscript, Pook and colleagues describe a new imputation pipeline for calling genotypes from whole-genome sequence data with very low read depth (0.5X). A major goal of the pipeline is to increase marker density though genotype-by-sequencing (GBS) when compared to genotype arrays, for the purposes of e.g. genomic prediction or association studies. Using sequence data from doubled haploid maize lines, the authors demonstrate lower imputation error rates with HBimpute when compared to BEAGLE. When these imputed markers are used for genomic prediction, the results are somewhat mixed—there is a reduction in predictive ability that the authors claim is due to the poorer quality of additional markers gained from the GBS approach.

While I do not doubt the pipeline by the authors produced lower imputation error in their dataset when compared to BEAGLE, I have concerns about some of the authors’ claims as well as with the novelty and broader applicability of the approach.

Average read depth is a useful shorthand to communicate the rigor of sequencing efforts across different platforms and pipelines. Here, the authors merge reads to claim a “virtual read depth” of 83X. This would be fine if they were interested in variants from a population or a species, but is definitely misleading when they are interested in the genotypes from specific lines and individuals. The claim that this depth of coverage enables structural variation (only CNVs) to be called is similarly misleading because they are called on what is effectively a population level when the method is aimed at describing genotypes at the individual level. I think the manuscript would be improved if the distinction between describing variation at the individual and the population level was made clearer.

The authors cite several approaches for imputation, but only benchmark their pipeline to BEAGLE and only with their dataset of doubled haploid maize lines. A custom pipeline for a specific dataset will typically produce better results than one off the shelf. To convincingly demonstrate better performance, the pipeline should be benchmarked against at least more than one existing method. Limitations from data availability may preclude the authors from testing the approach on additional datasets, but they should consider simulation strategies. Broadening the applicability of their approach would increase the appeal of the method to a wider audience.

Finally, while the goal of the new pipeline is to increase the number of available markers through GBS, array data is still treated as their gold standard. The error rates for imputed sequence are all reported as discordance with the genotyping array and genomic prediction is actually worse with the greater number of markers. The results would seem to indicate that array data is typically preferable than low depth whole-genome sequencing for the purposes that the authors are interested in. Heterozygote calls, for example, cannot be used from low depth sequencing while they are fine from array data.

Minor:

The authors rightly indicate that additional SNPs artificially inflate the power of GWAS, as measured by p-value, in ways that are difficult to compare. However, they proceed to simulate QTL and compare the proportion of true positive hits and use this as a metric of performance. It is not obvious to me how this strategy produces a comparable metric and this section could be explained a little better.

L193:

negligible should probably be used here and throughout the manuscript instead of “neglectable”

**Have all data underlying the figures and results presented in the manuscript been provided?**

Reviewer #1: Yes

Reviewer #2: Yes

Reviewer #3: **No: **As stated by the authors, only some of the data has been made available due to its private ownership by KWS Saat.

PLOS authors have the option to publish the peer review history of their article (what does this mean?). If published, this will include your full peer review and any attached files.

Reviewer #1: No

Reviewer #2: No

Reviewer #3: No

---

## [Decision Letter · Decision Letter 1]

31 Aug 2021

Dear Dr Pook,

Thank you very much for submitting your Research Article entitled 'Increasing calling accuracy, coverage, and read-depth in sequence data by the use of haplotype blocks' to PLOS Genetics.

The manuscript was evaluated at the editorial level and by independent peer reviewers. While two reviewers are now largely satisfied, reviewer 2 has substantial remaining concerns that that we ask you address in a revised manuscript and letter of response.  Also the point raised by reviewer 1 about availability of software is important.  While this is formally a "minor revision" decision, if your revisions take more than the 30 days suggested below that is not a problem.

We therefore ask you to modify the manuscript according to the review recommendations. Your revisions should address the specific points made by each reviewer.

[LINK]

Yours sincerely,

Jonathan Marchini

Associate Editor

PLOS Genetics

David Balding

Section Editor: Methods

PLOS Genetics

Reviewer's Responses to Questions

Reviewer #1: This is the revision of a manuscript presenting a method called HBImpute for estimating genotypes from low coverage (e.g. 0.5x) sequence data. The method is designed for the special case of samples with homozygous genotypes (double haploid lines) that arise in plant breeding. The method identifies haplotype blocks and clusters sequence reads from identical by descent haplotypes in each block. Sequence reads for a sample are augmented with sequence reads from other identical by descent haplotypes. The method gives an approximate 50% reduction in genotype error rates over a competing methods on an evaluation data set with 0.5x sequence coverage when compared to array genotypes. The HBImpute genotypes were used for association analysis and phenotype prediction, and yielded results that were similar to results obtained from SNP array data. The augmented sequence coverage appears to improve detection and calling of copy number variants.

The authors have addressed my previous comments. I have only two additional comments:

1) The response to the reviewers states that the software is freely available for academic research (“Use in academia is possible without restrictions”). This should also be stated in the published manuscript.

2) Line 479, “1.000 SNPs”. Do you mean 1000 or 1?

Reviewer #2: Thank you for your response to the reviewer comments. The quality of the manuscript improved by introducing other methods to the benchmark.

However, I still find the manuscript lacking of important information. In particular the accuracy comparison against STITCH, the second best method after the HB pipeline, should be more broadly expanded, to justify and show where the benefit of the HB pipeline resides over STITCH that is at least 5 times more computationally efficient then HB-array and 10 times more efficient than HB-seq (in both reported running time and memory), especially after seeing that they both show almost identical predictive power.

Major comments:

1. I appreciate the introduction of Figure 2. However, I honestly do not understand the author’s comment about avoiding to use the well-known and standard (dosage) imputation r^2. Verifying not only hard calls, but also dosages, is important when imputation is performed. A quantification of error rates stratified by REF and ALT calls for all the methods, would also be useful.

2. Connected to the previous comment, calibration of genotype posteriors for the HB pipeline seems also easy to check and important to verify that the introduction of the haplotype block (and therefore the merging of the reads) is sound.

3. The repeated statement of having a “read depth of 83X” is very misleading. To my understanding, this inflation of read depth that the authors claim is “artificial” is just for internal use of the pipeline and it is not seen (to that amount) in practice. The method is doing slightly better than other methods that work on pure 0.5x data, showing that the reported 83x is an inflated estimate.

4. The GWAS analyses is to me not convincing. The low sample size and no application of filtering (e.g. on MAF), lead to a questionable power and a small amount of error and bias in the genotype calls can produce many false positives.

5. Authors need provide the parameters they used to run all methods. The manuscript needs to significantly improve in terms of reproducibility.

Reviewer #3: The revisions by Pook et al. have substantially improved the manuscript. I appreciate the additional comparison to STITCH, the expanded discussion, and I now find the article suitable for publication.

**Have all data underlying the figures and results presented in the manuscript been provided?**

Reviewer #1: Yes

Reviewer #2: Yes

Reviewer #3: Yes

PLOS authors have the option to publish the peer review history of their article (what does this mean?). If published, this will include your full peer review and any attached files.

Reviewer #1: No

Reviewer #2: No

Reviewer #3: No

---

## [Decision Letter · Decision Letter 2]

13 Nov 2021

Dear Dr Pook,

We are pleased to inform you that your manuscript entitled "Increasing calling accuracy, coverage, and read-depth in sequence data by the use of haplotype blocks" has been editorially accepted for publication in PLOS Genetics. Congratulations!

Yours sincerely,

Jonathan Marchini

Associate Editor

PLOS Genetics

David Balding

Section Editor: Methods

PLOS Genetics

Comments from the reviewers (if applicable):

Reviewer #2: I still think that the claim "while the average read-depth is increased to 83X thus enabling the calling of copy number variation" reported in the abstract is misleading and should be rephrased to a more cautions statement.

However, the revision have improved the manuscript and the authors provided me reasonable answers to my comments. Therefore I think it is eligible to publication.

**Have all data underlying the figures and results presented in the manuscript been provided?**

Reviewer #2: Yes

PLOS authors have the option to publish the peer review history of their article (what does this mean?). If published, this will include your full peer review and any attached files.

Reviewer #2: No

**Data Deposition**

http://datadryad.org/submit?journalID=pgenetics&manu=PGENETICS-D-21-00034R2

**Press Queries**

---

## [Editor Report · Acceptance letter]

5 Dec 2021

PGENETICS-D-21-00034R2 

Increasing calling accuracy, coverage, and read-depth in sequence data by the use of haplotype blocks 

Dear Dr Pook, 

We are pleased to inform you that your manuscript entitled "Increasing calling accuracy, coverage, and read-depth in sequence data by the use of haplotype blocks" has been formally accepted for publication in PLOS Genetics! Your manuscript is now with our production department and you will be notified of the publication date in due course.

With kind regards,

Olena Szabo

PLOS Genetics

On behalf of:
